# Effects of Curved-Path Gait Training on Gait Ability in Middle-Aged Patients with Stroke: Protocol for a Randomized Controlled Trial

**DOI:** 10.3390/healthcare11121777

**Published:** 2023-06-16

**Authors:** Youngmi Jin, Yubin Lee, Seiyoun Park, Sangbin Lee, Chaegil Lim

**Affiliations:** 1Department of Special Education of Graduate School, Dankook University, Yongin 16890, Republic of Korea; jym790612@hanmail.net; 2Department of Health Science, Gachon University Graduate School, Incheon 21936, Republic of Korea; edbql@naver.com; 3Department of Physical Therapy, Wonkwang Health Science University, Iksan 54538, Republic of Korea; seiyoun9@hanmail.net; 4Department of Physical Therapy, Namseoul University, Cheonan 31020, Republic of Korea; 5Department of Physical Therapy, Gachon University, Incheon 21936, Republic of Korea

**Keywords:** stroke, curved path, stride, gait ability, proprioception

## Abstract

(1) Introduction: This study aimed to investigate the effects of curved-path stride gait training on the gait ability of patients with stroke. (2) Materials and Methods: Thirty patients with stroke were randomly assigned to curved-path stride gait training (*n* = 15) and general gait training groups (*n* = 15). Both groups underwent training for 30 min five times a week for 8 weeks. The gait ability of each was assessed using the Dynamic Gait Index (DGI), Timed-Up-and-Go (TUG) test, 10-meter walk test, and Figure-of-8 walk test (F8WT). (3) Results: The curved-path gait training group showed significant differences in the DGI, TUG test, 10-m walk test, and F8WT pre- versus post- intervention (*p* < 0.05). The general gait training group showed no significant difference in F8WT pre- versus post-intervention (*p* > 0.05). Additionally, there was a statistically significant intergroup difference in gait ability (*p* < 0.05). (4) Conclusions: Curved-path gait training resulted in greater improvement in gait ability than general gait training. Therefore, curved-path gait training can be a meaningful intervention for improving the gait ability of patients with stroke.

## 1. Introduction

Gait, an interaction between the body and central nervous system, is a complex technique involving intentional movements that quickly adapt to changing circumstances [1,2,3]. Gait has various forms, including straight- and curved-path strides. In a straight gait, body movements occur symmetrically, with the weight load and weight shifting balanced between the feet [4]. However, a curved-path gait requires kinetic asymmetry. A curved-path gait is accomplished by increasing the mediolateral ground reaction forces throughout the stance phase to propel the body in the desired direction. The lateral impulse is greater on the outside limb and lesser on the inside limb during a curved versus straight gait [5,6]. In addition, during curved-path walking, the mediolateral displacement of the body mass shifts inward toward the inner leg [7].

Patients with stroke show gait disorders resulting from damage to the motor and sensory pathways [8,9]. These patients show lower-limb asymmetry due to a lack of appropriate muscle contraction, which impedes their ability to assume a normal gait [10]. Patients with stroke fail to bear sufficient weight on the affected inner leg and adopt a reduced speed while walking along a curved path toward the affected side [11]. Thus, the kinematics and muscular components engaged during a curved-path gait toward the affected direction should be considered to improve its performance [12].

Common gait training attempts to achieve a normal range of motion in the joints and inhibit abnormal movements. In patients with stroke, gait training involves teaching weight loading and using sensory stimuli to move the load to the affected lower extremities [13]. Several studies have examined curved-path gait training. Kim et al. (2012) [14] found that eight-way curved-path gait training has a significant effect on balance and ambulation in patients with stroke. Additionally, Richard (2010) [15] reported that patients with stroke who undergo training that changes the center of gravity and rotates the pelvis show improved curved-path gait performance through improved adaptability of motor control on the affected side, which improves their ability to provide eccentric sensory information to the feet, move weight to the lower extremities, and support the body when walking along a curved path.

Moreover, to successfully assume a curved gait, the head and eyes must first move along the curved path [4,7]. In other words, the concentric sensory input of the neck and vestibular system, including the eye, provides information about movement and position, resulting in a normal rotational gait.

However, patients with stroke have limited information about their body position owing to poor joint stability and proprioceptive senses on the affected side, which reduces efficacy and efficiency during movement and reduces stride and gait speed during walking [16]. These patients also avoid weight loading and visually check the position of their feet on the affected side, thereby relying more on visual information [17]. Lamontagne et al. (2007) [18] suggested that to enhance the proprioceptive sensibility of patients with stroke, a shift in gaze and head position in the direction of the curve’s center should be performed, while ground reaction force, vestibular sense, and proprioceptive sense input to the affected foot are necessary. Gdowski et al. (2000) [19] reported that rotating the body along a curve activates sensory inputs to the neck and vestibular organs, stimulates the vestibular nucleus of the brainstem, and can contribute to leg muscle activation.

As shown above, a curved-path gait enhances the proprioception of patients with stroke, giving them a more efficient gait ability. Regarding the effect of curved-path walking, Kim et al. (2012) [14] reported a significant increase in static and dynamic balance in patients with stroke after 4 weeks of figure 8 versus straight-path walking. It is difficult to attribute this effect solely to the curved-path gait training because the figure 8 walking also included straight sections. Previous studies focused on directional change-related research but did not address the effects of complete rotational gait training. Therefore, this study aimed to compare the efficacy of complete curved-path gait training (without any straight sections) and general gait training at improving gait ability in middle-aged stroke patients with hemiplegia after an 8-week intervention. We also aimed to provide basic clinical data for the development of rotational walking training programs for patients with central nervous system lesions (e.g., stroke).

## 2. Materials and Methods

### 2.1. Participants

The study participants were recruited from the Gyeonggi K Hospital. A total of 34 middle-aged patients who were diagnosed with hemiplegia caused by middle cerebral artery stroke more than 6 months prior and agreed to participate in the study were included. Informed consent was obtained from all participants. The training period was 8 weeks long. The inclusion criteria were as follows: first stroke, ability to walk at least 6 m independently without assistance [20], Mini-Mental State Examination—Korean version score ≥24, willingness to participate in the study [14], no vision deficits or vestibular system abnormalities, and no orthopedic disease in the body or lower limbs [21]. The study was conducted in accordance with the guidelines of the Declaration of Helsinki and approved by the Institutional Review Board of the Gachon University (1044396-202103-HR-063-01) and Clinical Research Information Service (CRIS) (no. KCT0008171).

### 2.2. Procedures and Intervention

Of the 34 participants, 1 did not meet the inclusion criteria, and 3 were excluded for personal reasons; thus, a total of 30 participants were included in the study. To minimize bias, the 30 included patients were randomly assigned to the curved-path stride gait training (*n* = 15) and general gait training groups (*n* = 15). Both groups received gait training for 30 min five times a week for 8 weeks (Figure 1). The randomization was performed by a nonparticipating researcher using Microsoft Excel for Windows (Microsoft Corporation, Redmond, WA, USA), and each patient was randomly assigned a number between 1 and 30. Participants with even numbers were assigned to the curved-path stride gait training group, whereas those with odd numbers were assigned to the general gait training group. The assignment was recorded in a password-protected document by the researcher who performed the randomization. The participants were blinded to information regarding the group to which they did not belong, types and differences in the intervention programs, and variables being compared between the groups. All data were collected before and at the end of the 8-week intervention period by the same physical therapist who did not participate in the study. All walking times during each measurement were measured using a stopwatch, and all length measurements were performed using a ruler. The exercise programs were conducted under the supervision of a therapist. To prevent falls, safety protective equipment, such as corner protection and mattresses, were placed nearby. If the participants experienced dizziness during any exercise phase, they rested on a chair with a backrest for at least 1 min.

#### 2.2.1. Curved-Path Gait Training Program

The curved-path gait training program was created by adjusting the stride in the shape of a ladder on a circular gait path [22,23]. The circle path consisted of three circles: large (total length: 12.6 m; radius: 2 m; curvature: 0.5 m), medium (total length: 6.3 m; radius: 1 m; curvature: 1 m), and small (total length: 3.1 m; radius: 0.5 m; curvature: 2 m) (Figure 2). The outer curves of all circles are marked with dotted lines. A solid line was used to mark the inner curve, which was 60 cm away from the outer curve [22,23]. A ladder was marked between the two curves at the participants’ average stride (31 cm). A 1.7 cm wide black tape was used for all markings (Figure 2). The curved-path training program consisted of the following steps. First, the patients moved along the curved path to the right, sat in a chair, rested for 1 min, and then followed the path again to the left. As the curvature increased in the order of the large, medium, and small circles, the task became more challenging for the smaller circles. If a participant was unable to perform the task in a large circle (e.g., lost balance, needed continuous assistance, or had a slow walking speed), we did not proceed with the middle circle intervention. The same rule was applied to the middle and small circles.

The starting position of the training was first directed at the curve marked with white tape in the direction of the eyes and head. The participants were then asked to walk as close to the white tape as possible using one foot at a time. The participants then placed their feet on a black ladder. Before the participants stepped on the left foot, they placed their right foot next to the left foot to support the weight. In the first stage, the patients could use their gaze and head position to reorient themselves when they rotated along a curved path [7,24,25]. The curved-path training program consisted of a warm-up (5 min), main exercise (20 min), and cool-down (5 min) performed for 30 min five times a week for 8 weeks.

#### 2.2.2. General Gait Training Program

The general gait training program was a 45 m round trip training program performed for 30 min five times a week for 8 weeks. The gait training program consisted of a warm-up (5 min), main exercise (20 min), and cool-down (5 min). During this program, participants maintained a forward gaze, focusing on the front. If the participant adapted to the program without losing their balance and showed improved walking speed compared to the initial speed, the walking speed was gradually increased, and the assistance was reduced.

### 2.3. Outcome Measurements

#### 2.3.1. Ten-Meter Walk Test

The 10-meter walk test, which is commonly used to assess the gait ability of patients with neurological impairments [26,27,28], involves measuring the time required for a patient to walk 10 m in a straight line, excluding the acceleration and deceleration sections of 2 m each at the start and end. After a 10 m walkway was created between the two points, the walking time was measured [29]. It was conducted at the self-selected walking speed, and a total of 3 trials were made to obtain the average value, which was used as data [27,28]. The interrater reliability of this test is 0.87 [30]. For this study, it was measured three times, and the mean value was used for the analysis.

#### 2.3.2. Dynamic Gait Index

The Dynamic Gait Index (DGI) is a tool that was developed to assess an individual’s stability and risk of falling during walking [26,31,32,33]. The DGI consists of eight categories: marching on a flat surface, changing gait speed, gait with horizontal movements (rotation) of the head, gait with vertical movements (rotation) of the head, marching and turning on one’s own body axis (pivot), marching over an obstacle, bypassing obstacles, and marching up and down stairs [34]. Each item is scored on a 4-point scale of 0–3, with 0 representing the lowest level of function, and 3 representing the highest for a total possible score of 24 points [35]. A score ≥22 indicates that the patient can move safely, while a score ≤19 indicates a risk of falling. The test–retest reliability of the DGI was 0.96, while the interrater reliability was 0.96 [31].

#### 2.3.3. Figure-of-8 Walk Test

The Figure-of-8 walk test (F8WT) was used to assess curved walking ability [36,37,38,39]. In this test, the participants walked clockwise and counterclockwise around two cones. The measurable parameters for this test were time, number of steps, accuracy, and natural movement. The test begins with the patient walking between the two cones and stops when the patient returns to their starting position. We performed this test thrice, obtained average values, and used only the time and number of steps obtained in the analysis. Accuracy was indicated by a “yes” if the patient did not deviate from the cones and “no” if the patient deviated away from them. The natural movement item has a total score of 3 points, the subitems of which are given 0 points for difficulties related to pauses, hesitations, and changes in speed and 1 point if no difficulty is noted. The test–retest, intrarater, and interrater reliabilities were 0.98 each [38].

#### 2.3.4. Timed-Up-and-Go Test

The Timed-Up-and-Go (TUG) test is used to assess the mobility, balance, and motor ability of stroke patients [40,41]. In the TUG test, the time required for the participant to turn around a point 3 m away from the chair and return to his/her location was measured three times to obtain an average value. A time of <10 s indicated independent walking, <20 s indicated that some assistance is required, and >30 s indicated that considerable assistance is required for walking. The intrarater and interrater reliabilities of the TUG test were 0.99, 0.98, and 0.98, respectively [42].

### 2.4. Sample Size Estimation

This study used a single-blind randomized controlled trial design. G power 3.1.9.2 software (Heinrich Heine University Düsseldorf, Düsseldorf, Germany) [43] was used to determine the sample size [44]. An effect size of *d* = 1.34 was used to calculate the sample size based on the result of the F8WT (time and steps) in a prior pilot test [45]. Twenty-six participants were required to detect statistical significance when a clinically significant difference was observed between two independent means, with an effect size of *d* = 1.34, significance level of 0.05, and power of 0.90 [43]. An additional 30% quantity of patients was recruited to account for unanticipated dropouts.

### 2.5. Statistical Analysis

The statistical analysis was performed using the Statistical Package for the Social Sciences software (version 23.0; IBM, Armonk, NY, USA). All measured values are presented as mean and standard deviation. The Levene’s homogeneity of variance test was conducted to test the general characteristics and homogeneity of the participants before the experiment. A paired *t*-test was performed to analyze the differences between the dependent variables according to the measurement period (before and after the experiment), while an independent *t*-test was used to compare the effects of gait training on gait ability between the two groups.

The effect size was calculated to explain the arbitration effect more objectively. The significance level was set at *p* < 0.05.

## 3. Results

The DGI, TUG test, 10-meter walk test, and F8WT were used to measure participant gait ability. No significant intergroup differences were found (*p* > 0.05) in general characteristics (Table 1).

The dynamic gait ability results with curved-path versus general gait training in patients with stroke are shown in Table 2 and Table 3. The curved-path gait training group showed significant differences in all variables pre- versus post-training after 8 weeks, whereas the general gait training group showed no significant differences in the F8WT (speed and steps) (Table 2). The intergroup differences based on the differences in pre- and post-intervention after 8 weeks showed significant results for all variables (Table 3). The effect size was greater than 0.5 (Cohen’s *d* = 0.5) for DGI, F8WT speed, and F8WT steps.

## 4. Discussion

This study aimed to determine the effect of a curved-path gait training program on walking in patients with stroke. Significant improvements were observed in both curved-path gait training and general gait training pre- and post-intervention after the 8-week intervention; however, the curved-path gait training showed a significant improvement in gait ability compared to the general gait training group. Both training methods improved weight load, weight shifting, and speed changes on the affected side. It is important to note the significantly greater increase in walking speed observed after the intervention, particularly in the curved-path gait training group. This improvement in walking speed indicates enhanced muscle activation on the affected side, improved weight shifting and bearing, increased stance time on the affected side, and ultimately results in a more balanced walking pattern compared to general gait training. Eng et al. (2008) [46] also reported that curved-path gait training, in which eye and head movements occur first when walking, stimulates the affected side and shifts the body’s center of gravity, which can result in segmentation movements. Kwon et al. (2013) [47] reported that walking through gaze-oriented movements effectively improves gait ability. Based on these findings, patients with stroke require curved-path gait training, which includes changes in direction, as the body center is biased toward the unaffected side, the affected side is neglected, balance becomes difficult, and walking control decreases.

On the TUG test, the curved-path gait training group showed statistically significant improvements compared with the general gait training group. Consistent with the results of this study, Kim et al. (2012) [14] noted that curved-path gait training groups showed significant improvements in dynamic balance compared with general straight gait training groups. These results suggest that the concentric sensory information of the foot is improved by increasing the weight-shifting ability and weight load on both legs through curved-path gait training in patients with stroke.

The 10-meter walk test results for both groups showed statistically significant improvements in gait speed after training. The curved-path gait training group showed statistically significant improvements compared to the general walking training group. In a similar study, Lee et al. (2003) [13] reported that the greater the weight support on the affected foot, the faster the walking speed; specifically, if the affected foot was close to the curve in a curved-path gait, the weight support increased and the walking speed was improved. Therefore, the results of this study showed that curved-path gait training was more effective than general straight walking training in terms of weight support training on the affected side.

On the F8WT, the curved-path gait training group demonstrated improved gait speed and stride compared with the general gait training group. Similar to the results of this study, Kim et al. (2012) [14] reported that stroke patients who performed curved walking training in both directions on a figure-8-shaped track showed improved weight support and weight shifting in both legs. Ko et al. (2015) [48] also reported that exercise programs that improve proprioception on the affected side effectively improved gait ability and speed in stroke patients. Gdowski et al. (2000) [19] also noted that a rotating gait in a curved direction contributes to selective leg muscle activation by proprioceptive stimulation of the neck and vestibular nucleus of the brain. Therefore, walking along a curved path is believed to increase proprioceptive stimulation by changing gaze and head direction during walking, resulting in improved gait ability.

Chen et al. (2014) [12] reported that stroke patients require more time for turning compared to healthy individuals. This slow turning leads to poor balance ability and temporal gait asymmetry. The study also found that knee extensor muscle activation is dependent on speed and stroke patients had lower activation of the knee extensor muscles on the affected side compared to healthy individuals, regardless of the turning direction. This is related to the slower turning speed in patients with stroke. Bland et al. (2019) [49] stated that greater spatial variability and smaller temporal variability are associated with improved curved-path gait ability. Specifically, they highlighted the relationship between spatial variability and both step length and step width. Lee JH (2022) [50] investigated the spatiotemporal characteristics of the curvilinear gait compared to the rectilinear gait in hemiplegic stroke patients using surface electromyography. From a spatial motor module perspective, it was indicated that during the curvilinear gait, the weighting of knee extensor muscles decreased in the early stance phase of the gait cycle, suggesting difficulty in weight acceptance. In the early swing phase, a pattern of simultaneous activation of hip flexor muscles and knee extensor muscles was reported. In summary, during the curvilinear gait, this implies incomplete weight support on the affected side and an increase in swing time. Next, from a temporal motor module perspective, it was observed that during the curvilinear gait, the activation peak timing in the terminal stance phase was delayed, which was associated with slower gait speed.

Our study showed that the curved-path gait training improved overall gait ability compared to general gait training. The decreased number of steps observed in the F8WT after curve-path gait training may be related to an increase in step length and a decrease in step width. These changes in step length and width resulted in improved gait speed compared to the general gait training. The increased gait speed indicates an improvement in the activation of knee extensor muscles on the affected side, ultimately resulting in a longer stance time on the affected side and an increased swing phase time on the unaffected side. The prolonged swing time helps to compensate for the difference in stride length between both legs, leading to a more symmetrical movement between both sides. In other words, it is considered positive that the curved-path walking training has resulted in an increase in walking speed compared to general walking training, as it creates more symmetrical movements through spatial and temporal changes such as increased weight acceptance on the affected side, increased step length and swing time, and decreased step width.

This study had some limitations. First, it included only participants with chronic middle cerebral artery stroke. It would be necessary to verify the effectiveness of various stroke cases as well, as the remaining function and prognosis of patients may differ depending on the damaged blood vessels (e.g., anterior cerebral artery and posterior cerebral artery). Second, one-sided focus training on the affected side was not performed during gait training. Third, we failed to measure the relationship between vestibular sense and proprioception or between muscle activity and improved walking ability. Fourth, we compared only curved-path gait training and general gait groups; no control group was included. Although there may be ethical issues with including a group of patients with stroke who receive no intervention, it could serve as an important control group. Fifth, the study did not consider upper limb movements, such as the trunk and shoulder, in relation to the gait ability of patients with stroke.

## 5. Conclusions

This study investigated the effects of curved-path gait training on gait ability in stroke patients with hemiplegia. Curved-path gait training resulted in significantly improved gait ability compared with general gait training. This improvement ultimately results in a more balanced walking pattern compared to general gait training. These results suggest that curved-path gait training can be a meaningful intervention for improving the asymmetrical gait ability of stroke patients with hemiplegia. However, the neurological effects of this type of training have not been clearly defined. Further research is needed to investigate the effects of curved-path gait training on other neurological patients with gait impairment and other related neurological conditions.

## Figures and Tables

**Figure 1 healthcare-11-01777-f001:**
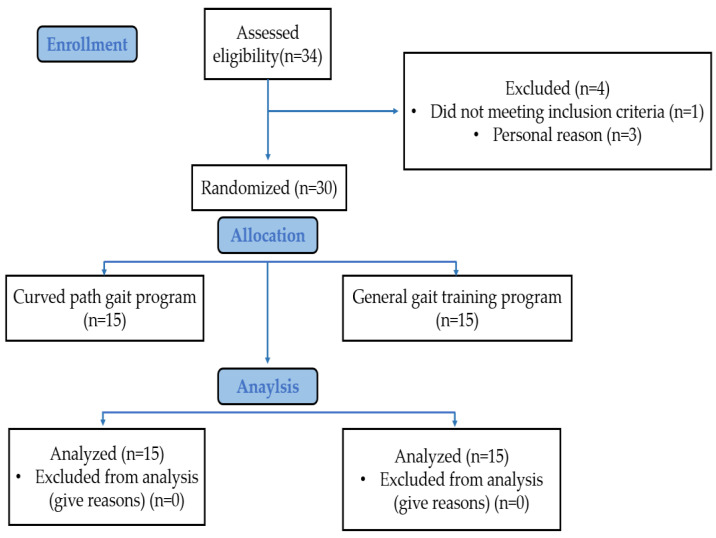
Experimental procedures.

**Figure 2 healthcare-11-01777-f002:**
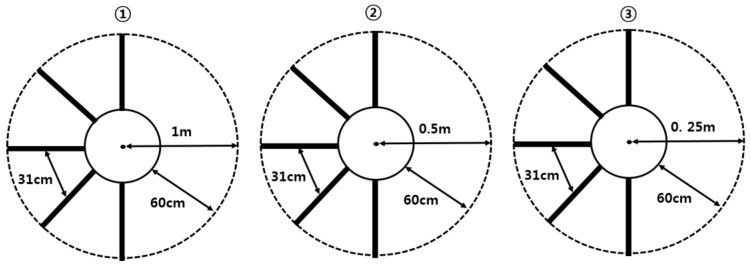
Curved-path gait training program: ① large circle; ② medium circle; ③ small circle.

**Table 1 healthcare-11-01777-t001:** General characteristics of the study participants and homogeneity test results (*N* = 30).

Category	EG (*n* = 15)	CG (*n* = 15)	*p*
Sex (male/female)	9/6	8/7	0.705 ^a^
Age (years)	54.38 ± 5.63	54.07 ± 7.60	0.905 ^b^
Height (cm)	162.84 ± 8.66	165.50 ± 9.17	0.448 ^b^
Weight (kg)	63.76 ± 10.06	64.42 ± 9.50	0.862 ^b^
Stroke type (hemorrhagic/embolism)	6/9	5/8	0.716 ^a^
Paretic side (right/left)	7/8	7/8	1.000 ^a^
Post-stroke duration (months)	6.23 ± 0.72	6.00 ± 0.67	0.401 ^b^
DGI (score)	16.40 ± 2.02	15.00 ± 3.54	0.104 ^b^
TUG (s)	19.37 ± 6.60	19.51 ± 7.47	0.504 ^b^
10 m walking (s)	18.10 ± 6.45	18.09 ± 8.51	0.242 ^b^
F8WT speed (s)	21.47 ± 7.36	21.48 ± 9.02	0.311 ^b^
F8WT steps (number)	23.80 ± 7.27	23.80 ± 9.19	0.260 ^b^

Data are expressed as mean ± SD or *n*. CG, control group (general gait training); DGI, Dynamic Gait Index; EG, experimental group (curved-path gait training); F8WT, Figure-of-8 walk test; TUG, Timed-Up-and-Go test. ^a^ Obtained using the χ^2^ test. ^b^ Obtained using an independent *t*-test.

**Table 2 healthcare-11-01777-t002:** Gait ability of patients with stroke by group before and after the 8-week intervention (*N* = 30).

	EG (*n* = 15)				CG (*n* = 15)			
Variable	Pre	Post	*p* *	E	Pre	Post	*p*	E
DGI (score)	16.400 ± 2.028	22.733 ± 1.624	0.000	0.886	15.000 ± 3.545	15.800 ± 3.342	0.001 *	0.728
TUG (s)	19.373 ± 6.603	15.246 ± 5.518	0.000	0.715	19.516 ± 7.473	19.033 ± 7.364	0.000 *	0.492
10 m walk (s)	18.106 ± 6.457	15.428 ± 6.227	0.001	0.547	18.097 ± 8.512	17.581 ± 8.306	0.001 *	0.276
F8WT speed (s)	21.473 ± 7.367	16.976 ± 6.656	0.000	0.770	21.484 ± 9.025	21.600 ± 8.941	0.337	0.631
F8WT steps (number)	23.800 ± 7.272	18.400 ± 6.936	0.000	0.820	23.800 ± 9.190	24.133 ± 9.030	0.096	0.705

Data are expressed as mean ± standard deviation. CG, control group (general gait training); DGI, dynamic gait index; E, effect size; EG, experimental group (curved-path gait training); F8WT, Figure-of-8 walk test; TUG, Timed-Up-and-Go test. * *p* < 0.05.

**Table 3 healthcare-11-01777-t003:** Differences in gait ability of patients with stroke by group before and after the 8-week intervention.

	EG (*n* = 15)	CG (*n* = 15)	
Variable	Difference (Post–Pre)	Difference (Post–Pre)	*p* *
DGI (score)	6.333 ± 2.350	0.800 ± 0.774	0.001
TUG test (s)	−4.127 ± 2.696	−0.483 ± 0.259	0.000
10 m walk test (s)	−2.678 ± 2.521	−0.516 ± 0.459	0.003
F8WT speed (s)	−4.496 ± 2.504	0.115 ± 0.449	0.000
F8WT steps (number)	−5.400 ± 2.613	0.333 ± 0.723	0.000

Data are expressed as mean ± standard deviation or *n* (%). CG, control group (general gait training); DGI, Dynamic Gait Index; E, effect size; EG, experimental group (curved-path gait training); F8WT, Figure-of-8 walk test; TUG, Timed-Up-and-Go test. * *p* < 0.05.

## Data Availability

The data used in this study are available upon request from the corresponding author. The data are not publicly available due to privacy or ethical considerations.

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
