# Peer review of "Effects of Curved-Path Gait Training on Gait Ability in Middle-Aged Patients with Stroke: Protocol for a Randomized Controlled Trial"

_healthcare, 2023, doi:10.3390/healthcare11121777_

Round 1

Reviewer 1 Report

This study deals with a topic of interest such as the treatment of mobility limitations in post-stroke patients. However, there are some aspects to point out. Some limitations, such as the very restrictive eligibility criteria, and the final n (despite the calculation of the sample size being correct) make it necessary to take the results of this work with caution when generalizing these results to the post stroke patient population. Also, it would have been interesting to measure the long-term benefits of treatment, not just the short-term benefits. It must be taken into account that patients will be aware of whether they are being disturbed when walking in a curve or in a line, despite the fact that they want to blind the evaluators. .

Author Response

Thank you for the detailed point-out. You are right. We will be reflected in future studies.

Reviewer 2 Report

The authors raise the issue of two kinds of gait training.

The topic of the manuscript is complementary to research on the impact of various forms of gait training. The subject validates the use of curved path gait training.  

To the authors:
The abstract section is not divided into introduction, materials and methods, results, and conclusions section.
Materials and methods:

Among the participants, What was the longest period of time that had passed from the diagnosis to the start of the project? 

The conclusions are consistent and appropriate. The references are appropriate.

Author Response

The abstract section is not divided into introduction, materials and methods, results, and conclusions section.

Thank you for the detailed point-out. We revise the sentence in abstract.

Materials and methods:

Among the participants, What was the longest period of time that had passed from the diagnosis to the start of the project? 

Thank you for the detailed point-out. We checked about 7 months post stroke duration.

Reviewer 3 Report

Thank you for your manuscript. It is interesting but you have some important questions to answer. Please see my comments below…

ABSTRACT

L18. “The DGI, TUG, 10-meter walk test, and F8WT significantly gait ability after gait training in both the experimental and control groups (p < .05).” – Improve sentence….

L24. “Instantaneous reflex stimulation was found to promote gait ability more in the curved-path stride group than in the general gait group.” – Your data do not support this conclusion…

INTRODUCTION

L29. “brain structures…” – I suggest “central nervous system”…

L31. “Gaits can include straight and curved-path stride gaits.” – I suggest: “Gait can include straight and curved-path strides.”…

L43. “Since this requires kinematics and muscular components during curved path gait toward the affected direction should be considered to improve curved path gait performance” – Kinematics is yielded…It is not a structure of the body or a capacity… Please, rephrase the sentence…

L46. “Common gait training attempts to achieve normal range of motion in the joints and inhibits abnormal movement.” – Reference?

L49. “Many studies have thus been conducted on curved-path gait training.” – First, an explanation on curved-path gait training is needed; second, which are the studies? References… Within the existing research, what is the need for the present study?

L62. “…which reduces efficiency when moving…” – I suggest: “…which reduces efficacy and efficiency when moving…”

L77. “…or elderly people with significantly reduced balance.” – This specific aim is not in line with your research…

METHODS

L83. “The inclusion criteria were as follows: participants diagnosed with middle cerebral  artery stroke that had an incidence period of more than six months” – However, table 1 shows that post-stroke duration are 4.23 and 4.00 months… I do not understand this…

L86. “…Mini-Mental State Examination (MMSE-K) score of 24 or higher…” – Why? Reference…

I do not understand why age is not an inclusion criterion…

L94. “Both groups received gait training for 30 min 5 times a week for 8 weeks…” – What is the theoretical rationale for this prescription? This rationale could be part of the Introduction section...

L95. “The baseline measurement of patient ability was performed prior to randomization.” – Why?

L101. “To maintain the blind…” – Improve…

L103. “During all exercise program, it was performed under the supervision of the therapist, and safety protective equipment such as corner protection and mattresses were placed nearby to prevent falls.” – Improve sentence. I suggest splitting it into two sentences…

L110. “The circle was 6.3 m in length with a 1-meter radius and curvature and a gap of 60 cm between the inner and outer curves. Improve sentence…

L115. “…first the patients moved along the curved path to the right, then they sat in a chair and rested for a minute, then they followed the path again to the left…” – What was the duration?

The explanation of the curved-path gait training program must be improved…

L124. “At first, the speed progressed slowly, and when adjusted, the speed gradually increased.” – What was the criteria for the increment?

L136. “The training started with the patient walking slowly and gradually increasing the speed, with less assistance as the patient adapted to the program.” – What was the criteria for the increment?

 Figure 2. It is not clear how the 60 cm and 31 cm were calculated…

L143. “…the walking time is measured.” – How? Stopwatch?

Figure-of-8 walk test… It is not clear that 3 trials were performed…

L191. “The independent t-test was used to test the homogeneity of the subjects.” – t-test is used to compare means…

L199. “Levene's homogeneity of variance test was conducted to test the general characteristics and homogeneity of the subjects before the experiment.” – This sentence must be in the statistical analysis section…

RESULTS

Data of the DGI, TUG, 10m walking, F8WT in Table 2 duplicate data showed in Table 3.

L206. “The curved-path gait training group was showed significant differences in all variables (Table 2). The general gait training group was showed significant differences in the DGI, TUG and 10-meter walk test (Table 2).” – Sentences must be improved…

Table 3. From the table and text is not perceptible what data means… title is also imperceptible…

Figure must be explained in the text…

DISCUSSION

L220. “Dynamic gait ability is the ability to maintain and adjust posture without swaying in various environments and is represented by sensory integration and motor control, such as proprioception, vestibular sense, and vision.” – Reference?

L228. “In particular, during curved-path gait training, the shift of weight along the line of sight, head rotation direction, and curve ratio increases concentric sensory stimulation on the affected foot, which is thought to improve walking ability.” – There are no data in this study to support this assertion. At most, speculate…

Another limitation is not having a group that does not do any training…

CONCLUSION

L274. “The comparison of the effects of curved-path gait training and general gait training on gait ability in this study showed significant effects in the DGI, TUG, and 10-meter walk tests in both groups.” – Improve sentence…

L280. “In addition, the curved-path gait training group showed statistically significant differences (p < .05).” Regarding what?

L282. “This is believed to improve gait ability by increasing the proprioception of the affected foot by controlling stride on a curved path in patients recovering from a stroke.” – Suggestion is different from conclusion…

No

Author Response

ABSTRACT

L18. “The DGI, TUG, 10-meter walk test, and F8WT significantly gait ability after gait training in both the experimental and control groups (p < .05).” – Improve sentence….

Thank you for the detailed point-out. We revise the sentence.

“The curved-path gait training group showed significant differences in the DGI, TUG, 10-meter walk test, and F8WT between pre- and post-intervention (p <.05). The general gait training group did not show a significant difference in F8WT (p >.05).”

L24. “Instantaneous reflex stimulation was found to promote gait ability more in the curved-path stride group than in the general gait group.” – Your data do not support this conclusion…

Thank you for the detailed point-out. We revise the sentence.

 “The curved-path gait training showed greater improvement in gait ability compared to the general gait training. Therefore, the curved-path gait training can be considered a meaningful intervention for improving the gait ability of stroke patients”

INTRODUCTION

L29. “brain structures…” – I suggest “central nervous system”…

Thank you for the detailed point-out. As you suggest, we have made the change to “cerebral system”.

L31. “Gaits can include straight and curved-path stride gaits.” – I suggest: “Gait can include straight and curved-path strides.”…

Thank you for the detailed point-out. As you suggest, we have made the change to “Gait can include straight and curved-path strides~”.

L43. “Since this requires kinematics and muscular components during curved path gait toward the affected direction should be considered to improve curved path gait performance” – Kinematics is yielded…It is not a structure of the body or a capacity… Please, rephrase the sentence…

Thank you for the detailed point-out. The previous study mentioned kinematics in relation to turning gait.

L46. “Common gait training attempts to achieve normal range of motion in the joints and inhibits abnormal movement.” – Reference?

Thank you for the detailed point-out. We believe that this sentence is a universally known and predictable fact, and therefore, there is no need to add any additional references.

L49. “Many studies have thus been conducted on curved-path gait training.” – First, an explanation on curved-path gait training is needed; second, which are the studies? References… Within the existing research, what is the need for the present study?

Thank you for the detailed point-out. We revised and added the contents

“Regarding the effect of curved-path walking, Kim et al. (2012) [14] reported a significant increase in static and dynamic balance in stroke patients after 4-weeks of figure of 8 walk compared to straight path walk. It is difficult to attribute this effect solely to curved-path gait training because the figure of 8 walk used in this study included straight sections as well. The previous studies have focused on directional change-related research and have not addressed the effects of complete rotational gait training. Therefore, the purpose of this study was to compare the efficacy of complete curved-path gait training (without any straight sections) and general gait training on gait ability of the stroke patients with hemiplegia after 8-weeks intervention.”

L62. “…which reduces efficiency when moving…” – I suggest: “…which reduces efficacy and efficiency when moving…”

Thank you for the detailed point-out. As you suggest, we have made the change to “which reduces efficacy and efficiency when moving~”.

L77. “…or elderly people with significantly reduced balance.” – This specific aim is not in line with your research…

= Thank you for the detailed point-out. We have deleted this sentence.

 METHODS

L83. “The inclusion criteria were as follows: participants diagnosed with middle cerebral  artery stroke that had an incidence period of more than six months” – However, table 1 shows that post-stroke duration are 4.23 and 4.00 months… I do not understand this…

= Thank you for the detailed point-out. We changed the number for post-stroke duration to 6 in Table 1.

L86. “…Mini-Mental State Examination (MMSE-K) score of 24 or higher…” – Why? Reference…

I do not understand why age is not an inclusion criterion…

= Thank you for the detailed point-out. We selected individuals with normal cognitive function, as indicated by a score of 24 higher on the MMSE-K, in order to accurately perform and understand the therapist`s instructions. We have added a reference for MMSE-K and revised the sentence to read, “middle-aged participants diagnosed with hemiplegia caused by middle cerebral artery stroke that had an incidence period of more than six months.”

L94. “Both groups received gait training for 30 min 5 times a week for 8 weeks…” – What is the theoretical rationale for this prescription? This rationale could be part of the Introduction section...

= Thank you for the detailed point-out. There is no separated evidence, but it was conducted in accordance with the regulations of the rehabilitation therapy system in Koera.

L95. “The baseline measurement of patient ability was performed prior to randomization.” – Why?

= Thank you for the detailed point-out. We have deleted this sentence.

L101. “To maintain the blind…” – Improve…

= Thank you for the detailed point-out. We revised the contents; “Randomization was performed by a non-participating researcher using Microsoft Excel for Windows software (Microsoft Corporation, Redmond, WA, USA), and each patient was randomly assigned a number between 1 and 30. Even numbers were assigned to the curved-path stride gait training group, while odd numbers were assigned to the general gait training group. The assignment was recorded in a password-protected document that only the researcher who performed the randomization. The participants were blinded to the information regarding the group they did not belong to types and differences in intervention programs and the variables being compared between the groups”.

L103. “During all exercise program, it was performed under the supervision of the therapist, and safety protective equipment such as corner protection and mattresses were placed nearby to prevent falls.” – Improve sentence. I suggest splitting it into two sentences…

= Thank you for the detailed point-out. We changed the contents “During all exercise program, it was performed under the supervision of the therapist. To prevent falls, the safety protective equipment such as corner protection and mattresses were placed nearby”

L110. “The circle was 6.3 m in length with a 1-meter radius and curvature and a gap of 60 cm between the inner and outer curves. Improve sentence…

= Thank you for the detailed point-out. We revised the sentences in 2.2.1 section.

“The circle path consisted of three different-sized circles; a large circle (total length: 12.6m, radius: 2m, curvature: 0.5m), a middle circle (total length: 6.3m, radius: 1m, curvature: 1m), and a small circle (total length: 3.1m, radius: 0.5m, curvature: 2m) (Figure 2). The outer curve of all circles were marked with a dotted line. A solid line was used to mark the inner curve, which was 60cm away from the outer curve [22,23]. A ladder was marked between the two curves, while maintaining the average stride of participants (31cm). A 1.7m wide black tape was used for all markings (Figure 2).”

L115. “…first the patients moved along the curved path to the right, then they sat in a chair and rested for a minute, then they followed the path again to the left…” – What was the duration?

= Thank you for the detailed point-out. The duration of each session varied among patients, so it is unknown. The curved-path gait training was conducted for 30 minutes.

The explanation of the curved-path gait training program must be improved…

= Thank you for the detailed point-out. We revised the sentences in 2.2.1 section.

“The circle path consisted of three different-sized circles; a large circle (total length: 12.6m, radius: 2m, curvature: 0.5m), a middle circle (total length: 6.3m, radius: 1m, curvature: 1m), and a small circle (total length: 3.1m, radius: 0.5m, curvature: 2m) (Figure 2). The outer curve of all circles were marked with a dotted line. A solid line was used to mark the inner curve, which was 60cm away from the outer curve [22,23]. A ladder was marked between the two curves, while maintaining the average stride of participants (31cm). A 1.7m wide black tape was used for all markings (Figure 2).”

“As the curvature increased in the order of the large, middle, and small circles, the task became more challenging with smaller circle. If a participant was unable to perform the task on the large circle (such as losing balance and needing continuous assistance or having a slow walking speed), we did not proceed to the middle circle intervention. The same rule was applied to the middle and small circles.“

L124. “At first, the speed progressed slowly, and when adjusted, the speed gradually increased.” – What was the criteria for the increment?

= Thank you for the detailed point-out. We deleted the contents

L136. “The training started with the patient walking slowly and gradually increasing the speed, with less assistance as the patient adapted to the program.” – What was the criteria for the increment?

= Thank you for the detailed point-out. We revised the content. “If the patient adapted to the program without losing balance and showed improvement in walking speed compared to the initial speed, the walking speed was gradually increased, and the assistance was reduced.”

 Figure 2. It is not clear how the 60 cm and 31 cm were calculated…

= Thank you for the detailed point-out. We revised and added sentences “A solid line was used to mark the inner curve, which was 60cm away from the outer curve [22,23]. A ladder was marked between the two curves, while maintaining the average stride of participants (31cm).”

L143. “…the walking time is measured.” – How? Stopwatch?

= Thank you for the detailed point-out. We added the sentence “All walking time during each measurement were measured using a stopwatch.” in the section 2.2 procedures and intervention.

Figure-of-8 walk test… It is not clear that 3 trials were performed…

= Thank you for the detailed point-out. We have revised sentence “For this study, the time and walk counts were repeated three times to obtain the average value” to “we performed this test three times and obtained the average values and use only the time and umber of steps obtained from this test.”

L191. “The independent t-test was used to test the homogeneity of the subjects.” – t-test is used to compare means…

= Thank you for the detailed point-out. We deleted the sentence and replaced it with “Levene`s homogeneity of variance test was conducted to test the general characteristics and homogeneity of the subjects before the experiment.” in the Results section.

L199. “Levene's homogeneity of variance test was conducted to test the general characteristics and homogeneity of the subjects before the experiment.” – This sentence must be in the statistical analysis section…

  = Thank you for the detailed point-out. We moved the sentence to Statistical analysis section

RESULTS

Data of the DGI, TUG, 10m walking, F8WT in Table 2 duplicate data showed in Table 3.

Thank you for the detailed point-out. Table 2 shows the differences within each group before and after the 8-weeks intervention, while Table 3 shows the differences between the groups based on the differences in values before and after the 8-weeks intervention.

L206. “The curved-path gait training group was showed significant differences in all variables (Table 2). The general gait training group was showed significant differences in the DGI, TUG and 10-meter walk test (Table 2).” – Sentences must be improved…

Thank you for the detailed point-out. We revised the contents “The curved-path gait training group showed significant differences in all variables between pre- and post-intervention after 8-weeks, but the general gait training group did not show any significant differences in F8WT (speed and step) (Table 2).”

Table 3. From the table and text is not perceptible what data means… title is also imperceptible…

 = Thank you for the detailed point-out. We revised the content. “The differences between the groups based on the difference values in pre- and post-intervention after 8-weeks showed significant results in all variables (Table 3).”

Figure must be explained in the text…

 = Thank you for the detailed point-out. We deleted the figure 3.

DISCUSSION

L220. “Dynamic gait ability is the ability to maintain and adjust posture without swaying in various environments and is represented by sensory integration and motor control, such as proprioception, vestibular sense, and vision.” – Reference?

Thank you for the detailed point-out. We deleted the sentence.

L228. “In particular, during curved-path gait training, the shift of weight along the line of sight, head rotation direction, and curve ratio increases concentric sensory stimulation on the affected foot, which is thought to improve walking ability.” – There are no data in this study to support this assertion. At most, speculate…

 Thank you for the detailed point-out. The effectiveness of current gait training has limitations in providing clear neurological evidence. Therefore, we explained that during curved-path gait training, certain factors such as weight shift along the line of sight, head rotation direction, and curve ratio increase sensory stimulation on the affected foot, which is believed to improve walking ability. We then supported this explanation by referencing previous studies on each of these factors.

Another limitation is not having a group that does not do any training…

 Thank you for the detailed point-out. We added the contents “Fourth, we compared only the curved-path gait training group and the general gait group without a group that did not receive any intervention. Although there may be ethical issues with having a group that did not receive any intervention, it could serve as an important control group.”

CONCLUSION

L274. “The comparison of the effects of curved-path gait training and general gait training on gait ability in this study showed significant effects in the DGI, TUG, and 10-meter walk tests in both groups.” – Improve sentence…

Thank you for the detailed point-out. We revised the conclusion section.

“This study was to investigate the effect of curved-path gait training on gait ability in stroke patients with hemiplegia. The curved path gait training showed a significant improvement in gait ability compared to general gait training. This improvement is attributed to the cooperative movement of the eyes and head during curved-path gait and the enhancement of concentric sensory input on affected limb, leading to improved gait ability such as stride length, walking speed. Therefore, the curved-path gait training can be considered a meaningful intervention for improving the asymmetrical gait ability of stroke patients with hemiplegia. However, the neurological effects of this training are not clearly defined. Further research is needed to investigated the effects of the curved-path gait training on other neurological patients with gait impairment and to conduct further neurological related research.”

L280. “In addition, the curved-path gait training group showed statistically significant differences (p < .05).” Regarding what?

Thank you for the detailed point-out. We deleted the content.

L282. “This is believed to improve gait ability by increasing the proprioception of the affected foot by controlling stride on a curved path in patients recovering from a stroke.” – Suggestion is different from conclusion…

Thank you for the detailed point-out. We revised the conclusion section.

“This study was to investigate the effect of curved-path gait training on gait ability in stroke patients with hemiplegia. The curved path gait training showed a significant improvement in gait ability compared to general gait training. This improvement is attributed to the cooperative movement of the eyes and head during curved-path gait and the enhancement of concentric sensory input on affected limb, leading to improved gait ability such as stride length, walking speed. Therefore, the curved-path gait training can be considered a meaningful intervention for improving the asymmetrical gait ability of stroke patients with hemiplegia. However, the neurological effects of this training are not clearly defined. Further research is needed to investigated the effects of the curved-path gait training on other neurological patients with gait impairment and to conduct further neurological related research.”

Reviewer 4 Report

This study aims to compare the efficacy of curved-path gait training with normal gait training in the walking ability of post-stroke patients. It is a randomized clinical trial, assessor-blinded, with 30 patients (15 curved-path training and 15 straight-path gait training). The training was 30 minutes, 5 times per week, for 8 weeks.  The experimental group showed superiority in improving gait ability when compared to the control group.

The study was well-designed and well-conducted. However, it needs changes in the writing of the manuscript.

Please see specific comments below:

General

It is interesting to standardize the terms "effect" and "efficacy" in the title and when the study objective is mentioned (introduction, discussion, and conclusion).

Title

Insert the type of study.

Abstract

Lines 18-19. This sentence should be rewritten.

Lines 22-25. This information was not found in this study. Insert information and clinical implications of the results found in this study.

Introduction

Line 31. “Gaits can include straight and curved-path stride gaits”. There are other types of gait, forward and backward, among others. It would be interesting to rephrase the sentence.

Line 32. “with weight load and weight movement balanced between both feet”. What does “weight movement” mean?

Line 41. “making it difficult for them a normal gait”. Something is missing in this sentence.

Line 43-45. Something is missing in this sentence.

Authors affirm: “Many studies have thus been conducted on curved-path gait training”. What is different about this study compared to other studies on the topic? Or what are the methodological flaws of other studies that justify carrying out this one?

Could the primary outcome be included in the study objective?

Materials and Methods

Insert a study type topic. It is interesting to follow CONSORT (Consolidated Standards of Reporting Trials) guidelines for manuscript writing.

Line 80. 34 patients. The authors say that one did not meet the inclusion criteria, and another did not consent to the study. As such, these two were not included and should not count as persons included. Please explain the reason for excluding or not including the other two. For the text to be more faithful to what happened, it would be interesting to state that 34 people were contacted, and of these, 30 were included.

Line 83-84. These criteria have already been mentioned in the included population, they do not need to be repeated.

Inclusion criteria: was it the first stroke? unilateral?

It is interesting to insert the reference to the MMSE-K cut-off score.

Does the study have a clinical trial registration?

Despite the description of blinding the patients, it is unlikely that the participants did not know whether they were walking on a straight or curved path.

Statistical analysis. The chi-square test was also used.

Results

The mean post-stroke months for both groups is around 4 months. How is this possible if only patients six months after the stroke were included?

In table 3 the results of the groups are exchanged.

Figure 3 is unnecessary. The results are already shown in Tables 2 and 3.

Discussion

In general, the discussion needs to be rewritten, looking for justifications for the results found and clinical implications of the results (for patients and therapists). Comparison with the results of other similar studies is important, but should not be the basis of the discussion.

Lines 220-224 can be deleted. This information is already in the introduction.

Lines 225-227 The results can be more detailed at the beginning of the discussion. The term “dynamic gait regulation does not characterize the outcomes evaluated in this study.

Lines 231-232 This information has already been mentioned in the discussion.

Paragraphs 2, 3, and 4 of the discussion begin with the results of specific tests. However, throughout the paragraphs, the results of the cited studies do not seem to be related to the mentioned tests.

Conclusion

The conclusion must be rewritten to answer the objective of the study. The objective is to evaluate the efficacy of curved-path training. Therefore, only the curved-path training effect and its difference from the control training should be shown. There was an improvement in the parameters of the evaluated tests and this improvement was superior to the straight-path training. It is not advisable to insert p values. It is interesting to insert a clinical implication for the patient or the therapist.

The introduction is hard to read for me. Many sentences seem incomplete, with missing elements (terms, words, and proper punctuation). Maybe it needs an English review.

Author Response

General

It is interesting to standardize the terms "effect" and "efficacy" in the title and when the study objective is mentioned (introduction, discussion, and conclusion).

 Thank you for the detailed point-out.

Title

Insert the type of study.

  Thank you for the detailed point-out. We revise the sentence “Effects of Curved-Path Gait Training on Gait Ability in Patients with Stroke: A Protocol for a Randomized Controlled Trial”

Abstract

Lines 18-19. This sentence should be rewritten.

Thank you for the detailed point-out. We revise the sentence.

“The curved-path gait training group showed significant differences in the DGI, TUG, 10-meter walk test, and F8WT between pre- and post-intervention (p <.05). The general gait training group did not show a significant difference in F8WT (p >.05).”

Lines 22-25. This information was not found in this study. Insert information and clinical implications of the results found in this study.

Thank you for the detailed point-out. We revise the sentence.

“The curved-path gait training showed greater improvement in gait ability compared to the general gait training. Therefore, the curved-path gait training can be considered a meaningful intervention for improving the gait ability of stroke patients”

Introduction

Line 31. “Gaits can include straight and curved-path stride gaits”. There are other types of gait, forward and backward, among others. It would be interesting to rephrase the sentence.

Thank you for the detailed point-out. We revise the sentence.

“Gait has various forms, among which are straight and curved-path stride.”

Line 32. “with weight load and weight movement balanced between both feet”. What does “weight movement” mean?

Thank you for the detailed point-out. We revise “weight movement” to “weight shifting”

Line 41. “making it difficult for them a normal gait”. Something is missing in this sentence.

Thank you for the detailed point-out. We revised the sentence. “These patients show asymmetry in their lower limbs due to a lack of appropriate muscle contraction, which impedes their ability to normal gait”

Line 43-45. Something is missing in this sentence.

Thank you for the detailed point-out. We confirmed the context of “Patients with stroke show gait disorders resulting from damage to the motor and sensory pathways [8,9]. These patients show asymmetry in their lower limbs due to a lack of appropriate muscle contraction, which impedes their ability to normal gait [10]. Patient with stroke failed to bear the weight sufficiently in the affected inner leg and reduced speed while curved path gait toward the affected side [11]. Since this requires kinematics and muscular components during curved path gait toward the affected direction should be considered to improve curved path gait performance” and no problem understanding it.

Authors affirm: “Many studies have thus been conducted on curved-path gait training”. What is different about this study compared to other studies on the topic? Or what are the methodological flaws of other studies that justify carrying out this one?

Could the primary outcome be included in the study objective?

Thank you for the detailed point-out. We revised and added the contents

“Regarding the effect of curved-path walking, Kim et al. (2012) [14] reported a significant increase in static and dynamic balance in stroke patients after 4-weeks of figure of 8 walk compared to straight path walk. It is difficult to attribute this effect solely to curved-path gait training because the figure of 8 walk used in this study included straight sections as well. The previous studies have focused on directional change-related research and have not addressed the effects of complete rotational gait training. Therefore, the purpose of this study was to compare the efficacy of complete curved-path gait training (without any straight sections) and general gait training on gait ability of the stroke patients with hemiplegia after 8-weeks intervention”

The introduction is hard to read for me. Many sentences seem incomplete, with missing elements (terms, words, and proper punctuation). Maybe it needs an English review.

Thank you for the detailed point-out. We have overall revised.

Materials and Methods

Insert a study type topic. It is interesting to follow CONSORT (Consolidated Standards of Reporting Trials) guidelines for manuscript writing.

Thank you for the detailed point-out. We added the content in 2.2 Procedure and intervention section. “This study was a pretest-posttest randomized control group design.”

Line 80. 34 patients. The authors say that one did not meet the inclusion criteria, and another did not consent to the study. As such, these two were not included and should not count as persons included. Please explain the reason for excluding or not including the other two. For the text to be more faithful to what happened, it would be interesting to state that 34 people were contacted, and of these, 30 were included.

Thank you for the detailed point-out. We added the sentence “Out of the 34 participants, one person did not meet the inclusion criteria and four persons were excluded due to personal reasons, resulting in a total of 30 participants who were included in the study” to the 2.2 Procedures and intervention section.

Line 83-84. These criteria have already been mentioned in the included population, they do not need to be repeated.

Thank you for the detailed point-out. We deleted this sentence.

Inclusion criteria: was it the first stroke? unilateral?

Thank you for the detailed point-out. We revised the sentence “A total of 34 middle-aged patients who were diagnosed with hemiplegia caused by middle cerebral artery stroke more than six months~”.

It is interesting to insert the reference to the MMSE-K cut-off score.

Thank you for the detailed point-out. We added the reference about MMSE-K.

Does the study have a clinical trial registration?

Thank you for the detailed point-out. It was approved by the Institutional Review Board of Gachon University (1044396-202103-HR-063-01) and the CRIS system (Number KCT0008171) in 2.1 Participants.

Despite the description of blinding the patients, it is unlikely that the participants did not know whether they were walking on a straight or curved path.

Thank you for the detailed point-out. We added content in 2.2 Procedures and intervention section “The patients were blinded to the information regarding the group they did not belong to types and differences in intervention programs and the variables being compared between the groups”.

Statistical analysis. The chi-square test was also used.

 Thank you for the detailed point-out. We are not use the chi-square test

Results

The mean post-stroke months for both groups is around 4 months. How is this possible if only patients six months after the stroke were included?

  = Thank you for the detailed point-out. We revised the post-stroke duration in Table 1.

In table 3 the results of the groups are exchanged.

  = Thank you for the detailed point-out. We revised the Table 3.

Figure 3 is unnecessary. The results are already shown in Tables 2 and 3.

= Thank you for the detailed point-out. We deleted the figure 3.

Discussion

In general, the discussion needs to be rewritten, looking for justifications for the results found and clinical implications of the results (for patients and therapists). Comparison with the results of other similar studies is important, but should not be the basis of the discussion.

Thank you for the detailed point-out. The effectiveness of current gait training has limitations in providing clear neurological evidence. Therefore, we explained that during curved-path gait training, certain factors such as weight shift along the line of sight, head rotation direction, and curve ratio increase sensory stimulation on the affected foot, which is believed to improve walking ability. We then supported this explanation by referencing previous studies on each of these factors.

Lines 220-224 can be deleted. This information is already in the introduction.

 Thank you for the detailed point-out. We deleted the contents.

Lines 225-227 The results can be more detailed at the beginning of the discussion.

Thank you for the detailed point-out. We have deleted the sentence and made the following modifications. “As a result, the significant improvements were observed in both curved-path gait training and general gait training in pre- and post-intervention after 8-weeks but, in the comparison between the groups, the curved-path gait training showed a significant improvement in gait ability compared to general gait training. This suggests that both training methods improved weight load, weight shifting, and speed changes on the affected side.”

The term “dynamic gait regulation does not characterize the outcomes evaluated in this study.

Thank you for the detailed point-out. We revised “dynamic gait regulation” to “dynamic gait ability”.

Lines 231-232 This information has already been mentioned in the discussion.

Thank you for the detailed point-out. We deleted the contents.

Paragraphs 2, 3, and 4 of the discussion begin with the results of specific tests. However, throughout the paragraphs, the results of the cited studies do not seem to be related to the mentioned tests.

 Thank you for the detailed point-out. The effectiveness of current gait training has limitations in providing clear neurological evidence. Therefore, we explained that during curved-path gait training, certain factors such as weight shift along the line of sight, head rotation direction, and curve ratio increase sensory stimulation on the affected foot, which is believed to improve walking ability. We then supported this explanation by referencing previous studies on each of these factors.

Conclusion

The conclusion must be rewritten to answer the objective of the study. The objective is to evaluate the efficacy of curved-path training. Therefore, only the curved-path training effect and its difference from the control training should be shown. There was an improvement in the parameters of the evaluated tests and this improvement was superior to the straight-path training. It is not advisable to insert p values. It is interesting to insert a clinical implication for the patient or the therapist.

Thank you for the detailed point-out. We revised the conclusion section.

“This study was to investigate the effect of curved-path gait training on gait ability in stroke patients with hemiplegia. The curved path gait training showed a significant improvement in gait ability compared to general gait training. This improvement is attributed to the cooperative movement of the eyes and head during curved-path gait and the enhancement of concentric sensory input on affected limb, leading to improved gait ability such as stride length, walking speed. Therefore, the curved-path gait training can be considered a meaningful intervention for improving the asymmetrical gait ability of stroke patients with hemiplegia. However, the neurological effects of this training are not clearly defined. Further research is needed to investigated the effects of the curved-path gait training on other neurological patients with gait impairment and to conduct further neurological related research.”

Reviewer 5 Report

Materials and methods

1. 34 patients were enrolled in the study but 4 patients were excluded. The reasons for exclusion should be stated in the methods.

2. It is not appropriate to say that patients who were middle cerebral artery stroke.

A middle cerebral artery stroke is a patient stroke caused by primary middle cerebral artery occlusion. Most patients with middle cerebral artery occlusion have hemiplegia and are dependent. The study included patients with hemorrhage stroke and ischemic stroke and most patients with mild stroke. It is not appropriate to use the term middle cerebral artery stroke.

3. Does informed consent obtained have to be stated for all patients?

4. Line 99 to 101, “The patients were blinded about their group's information. And they could not consistently distinguish between curve-path stride gait training and general gait training.”

Which information was blinded and how to blind their group's information?

Is there a reason people couldn't distinguish general gait training from curve-path stride training?

5. Figure 1.

No patient loss following-up, information on loss follow-up is not necessary.

Author Response

Materials and methods

  1. 34 patients were enrolled in the study but 4 patients were excluded. The reasons for exclusion should be stated in the methods.

 Thank you for the detailed point-out. We added the sentence “Out of the 34 participants, one person did not meet the inclusion criteria and four persons were excluded due to personal reasons, resulting in a total of 30 participants who were included in the study” to the 2.2 Procedures and intervention section.

  1. It is not appropriate to say that patients who were middle cerebral artery stroke.

A middle cerebral artery stroke is a patient stroke caused by primary middle cerebral artery occlusion. Most patients with middle cerebral artery occlusion have hemiplegia and are dependent. The study included patients with hemorrhage stroke and ischemic stroke and most patients with mild stroke. It is not appropriate to use the term middle cerebral artery stroke.

 Thank you for the detailed point-out. We thought that it is appropriate to specify MCA clearly, as there are differences in the severity of impairments depending on the damaged blood vessel.

  1. Does informed consent obtained have to be stated for all patients?

Thank you for the detailed point-out. We added the sentence “Informed consent was obtained from all participants prior to the study” in 2.1 participants.

  1. Line 99 to 101, “The patients were blinded about their group's information. And they could not consistently distinguish between curve-path stride gait training and general gait training.”

Which information was blinded and how to blind their group's information?

Is there a reason people couldn't distinguish general gait training from curve-path stride training?

 Thank you for the detailed point-out. We revised the contents; “Randomization was performed by a non-participating researcher using Microsoft Excel for Windows software (Microsoft Corporation, Redmond, WA, USA), and each patient was randomly assigned a number between 1 and 30. Even numbers were assigned to the curved-path stride gait training group, while odd numbers were assigned to the general gait training group. The assignment was recorded in a password-protected document that only the researcher who performed the randomization. The participants were blinded to the information regarding the group they did not belong to types and differences in intervention programs and the variables being compared between the groups”.

  1. Figure 1. No patient loss following-up, information on loss follow-up is not necessary.

Thank you for the detailed point-out. We revised the Figure 1.

Reviewer 6 Report

1. Please indicated that how many doctors (inspector) were included in the study, and how to solve if different opinions arise during function examinaiton.

Whether the doctors (inspector) is consistent before and after followed up.

2. Limitation of this study should be added.

Minor editing of English language required.

Author Response

  1. Please indicated that how many doctors (inspector) were included in the study, and how to solve if different opinions arise during function examinaiton.

Whether the doctors (inspector) is consistent before and after followed up.

Thank you for the detailed point-out. “All data were measured by the same blind physical therapist before and at the end of the 8-weeks intervention period” is specified in 2.2 Procedure and intervention section.

  1. Limitation of this study should be added.

Thank you for the detailed point-out. We added the content. “Fourth, we compared only the curved-path gait training group and the general gait group without a group that did not receive any intervention. Although there may be ethical issues with having a group that did not receive any intervention, it could serve as an important control group.”

Minor editing of English language required.

Thank you for the detailed point-out. We have edited the English language overall.

Reviewer 7 Report

Manuscript “Effects of Curved-Path Gait Training on Gait Ability in Patients with Stroke” described a curved-path stride gait training program to rehabilitate the gait ability in stroke patients. The results showed that the patients subjected to curved-path stride gait training program recovered better in terms of gait ability than the control group subjected to the general gait training program assessed with several different gait tests. In general, the experiments were carefully designed and performed, and the results were reliable based on the data presented in the manuscript. I only have some minor issues that need to be considered or addressed.

1.     The English language needs to be improved. There are many places with typos, misspellings and grammatic errors. Here I just list a few examples:

1) Page 1, line 20, “F8W” should be “F8WT”.

2) Page 1, line36, delete a “be”.

3) Page 1, line 41, “…making it difficult for them a normal gait”. Please rephrase.

4) Page 2, line 85, “with” should be “had”.

5) Page 2, line 42, “delete “who”.

6) Page 6, Table 1 footnotes: “Time Up and Go Test” should be “Timed up and Go Test”.

7) Page 6, line 210, “group” should be “groups”.

2.     Some sentences/phrases are hard to understand. Here are some examples:

1)     Page 2, line 57, “head-and-eye turnover”. Please explain a bit detailed.

2)     Page 2, line 95, “patient ability”. What kind of ability? Please explain.

3)     Page 3, line 103, “blind physical therapist”. The therapist shouldn’t be “blind”. Please rewrite.

4)     Page 3, line 222, “…himself when he…”, please use neutral gender expression because the study included both males and females.

3.     Figure 2. When look at the figure and read the text in page 3 it is difficult to find the labels in the figure that mentioned in the text, e.g., where is the white tape? Please improve the figure so that it becomes visually easy to understand.

4.     Table 2, The “*” shouldn’t be placed after “P” since not all P values are < 0.05. The “*” can just be deleted without affecting the meaning.

5.     Table 2, the statistical analyses for the CG: The data for “pre” and “post” in the first three tests differ not too much, but the P value are very small. Please check whether there are some mistakes.

6.     Table 3 and Figure 3 do not fit. I believe in Table 3, EG and CG were reversed. Please check and correct.

7.     The last part of the Abstract is too detailed, whose information cannot be obtained from the text in the first part of the Abstract. This is the content: ”… when shifting weight in the left, right, forward, and backward directions and supporting weight during a 360° gait. Instantaneous reflex stimulation…”. The information of ”the left, right, forward, and backward directions” and ” instantaneous reflex stimulation” cannot be seen from the text in the abstract. Please consider revising it and make the description more general.

8.     Conclusions read like a summary. Please consider rewriting it by avoiding the redundancy (e.g., the P values) and emphasizing the significance of the results.

The English of the manuscript needs to be improved. Selected points were listed in the Comments and Suggestions for Authors (point 1 and 2). I strongly suggest that the author find a native English speaker to thoroughly read and edit the manuscript.

Author Response

  1. The English language needs to be improved. There are many places with typos, misspellings and grammatic errors. Here I just list a few examples:

Thank you for the detailed point-out. We have overall revised

1) Page 1, line 20, “F8W” should be “F8WT”.

2) Page 1, line36, delete a “be”.

3) Page 1, line 41, “…making it difficult for them a normal gait”. Please rephrase.

4) Page 2, line 85, “with” should be “had”.

5) Page 2, line 42, “delete “who”.

6) Page 6, Table 1 footnotes: “Time Up and Go Test” should be “Timed up and Go Test”.

7) Page 6, line 210, “group” should be “groups”.

  1. Some sentences/phrases are hard to understand. Here are some examples:

Thank you for the detailed point-out. We have overall revised

1)     Page 2, line 57, “head-and-eye turnover”. Please explain a bit detailed.

2)     Page 2, line 95, “patient ability”. What kind of ability? Please explain.

3)     Page 3, line 103, “blind physical therapist”. The therapist shouldn’t be “blind”. Please rewrite.

4)     Page 3, line 222, “…himself when he…”, please use neutral gender expression because the study included both males and females.

  1. Figure 2. When look at the figure and read the text in page 3 it is difficult to find the labels in the figure that mentioned in the text, e.g., where is the white tape? Please improve the figure so that it becomes visually easy to understand.

Thank you for the detailed point-out. We revised the curved-path gait training program section and Figure 2.

  1. Table 2, The “*” shouldn’t be placed after “P” since not all P values are < 0.05. The “*” can just be deleted without affecting the meaning.

Thank you for the detailed point-out. We revised the Table 2

5.

Thank you for the detailed point-out. We checked the data but were OK.

  1. Table 3 and Figure 3 do not fit. I believe in Table 3, EG and CG were reversed. Please check and correct.

Thank you for the detailed point-out. We revised the Table 3 and deleted the Figure 3.

  1. The last part of the Abstract is too detailed, whose information cannot be obtained from the text in the first part of the Abstract. This is the content: ”… when shifting weight in the left, right, forward, and backward directions and supporting weight during a 360° gait. Instantaneous reflex stimulation…”. The information of ”the left, right, forward, and backward directions” and ” instantaneous reflex stimulation” cannot be seen from the text in the abstract. Please consider revising it and make the description more general.

Thank you for the detailed point-out. We have overall revised

  1. Conclusions read like a summary. Please consider rewriting it by avoiding the redundancy (e.g., the P values) and emphasizing the significance of the results.

Thank you for the detailed point-out. We have overall revised the Conclusion section

The English of the manuscript needs to be improved. Selected points were listed in the Comments and Suggestions for Authors (point 1 and 2). I strongly suggest that the author find a native English speaker to thoroughly read and edit the manuscript.

Thank you for the detailed point-out. We have overall revised

Round 2

Reviewer 3 Report

Thank you for the improvement of your manuscript. Please see my comments below…

According to the defined inclusion criteria, I suggest to change the title of the study for: “Effects of Curved-Path Gait Training on Gait Ability in middle-age Patients with Stroke: Protocol for a Randomized Controlled Trial”

I also suggest changing the aim for: “Therefore, this study aimed to compare the efficacy of complete curved-path gait training (without any straight sections) and general gait training at improving gait ability in middle-age stroke patients with hemiplegia after an 8-week intervention.”

DISCUSSION

L252. “In particular, during curved-path gait training, the shift in weight along the line of sight, head rotation direction, and curve ratio increases the concentric sensory stimulation of the affected foot, improving walking ability.” – I reaffirm what I have referred in my previous revision: there are no data in this study to support this assertion; at most, speculate…

CONCLUSION

L339. “This improvement is attributed to the cooperative movement of the eyes and head during the curved-path gait and the enhancement of concentric sensory input to the affected limb, leading to improved gait abilities, such as stride length and walking speed.” – I reaffirm what I have referred in my previous revision: suggestion is different from conclusion…

Author Response

According to the defined inclusion criteria, I suggest to change the title of the study for: “Effects of Curved-Path Gait Training on Gait Ability in middle-age Patients with Stroke: Protocol for a Randomized Controlled Trial”

= Thank you for the detailed point-out. We have made the modifications according to your suggestion.

I also suggest changing the aim for: “Therefore, this study aimed to compare the efficacy of complete curved-path gait training (without any straight sections) and general gait training at improving gait ability in middle-age stroke patients with hemiplegia after an 8-week intervention.”

= Thank you for the detailed point-out. We have made the modifications according to your suggestion.

DISCUSSION

L252. “In particular, during curved-path gait training, the shift in weight along the line of sight, head rotation direction, and curve ratio increases the concentric sensory stimulation of the affected foot, improving walking ability.” – I reaffirm what I have referred in my previous revision: there are no data in this study to support this assertion; at most, speculate…

 = Thank you for the detailed point-out. We revised the sentence.

“It is important to note the significantly greater increase in walking speed observed after the intervention, particularly in the curved-path gait training group. This improvement in walking speed indicates enhanced muscle activation on the affected side, improved weight shifting and bearing, increased stance time on the affected side, and ultimately results in a more balanced walking pattern compared to general gait training.”

CONCLUSION

L339. “This improvement is attributed to the cooperative movement of the eyes and head during the curved-path gait and the enhancement of concentric sensory input to the affected limb, leading to improved gait abilities, such as stride length and walking speed.” – I reaffirm what I have referred in my previous revision: suggestion is different from conclusion…

= Thank you for the detailed point-out. We revised the sentence.

“This improvement indicates ultimately results in a more balanced walking pattern compared to general gait training.”
